# Improving physical function of patients following intensive care unit admission (EMPRESS): protocol of a randomised controlled feasibility trial

Rebecca Cusack [ORCID],[1,2] Andrew Bates [ORCID],[1] Kay Mitchell [ORCID],[1] Zoe van Willigen,[3] Linda Denehy [ORCID],[4,5] Nicholas Hart [ORCID],[6,7] Ahilanandan Dushianthan [ORCID],[1,2] Isabel Reading [ORCID],[8] Maria Chorozoglou,[8] Gordon Sturmey,[9] Iain Davey,[9] Michael Grocott [ORCID] [1,2,8]

**Correspondence to**
Dr Rebecca Cusack;
R.Cusack@soton.ac.uk

## ABSTRACT

**Introduction** Physical rehabilitation delivered early following admission to the intensive care unit (ICU) has the potential to improve short-term and long-term outcomes. The use of supine cycling together with other rehabilitation techniques has potential as a method of introducing rehabilitation earlier in the patient journey. The aim of the study is to determine the feasibility of delivering the designed protocol of a randomised clinical trial comparing a protocolised early rehabilitation programme including cycling with usual care. This feasibility study will inform a larger multicentre study.

**Methods and analysis** 90 acute care medical patients from two mixed medical–surgical ICUs will be recruited. We will include ventilated patients within 72 hours of initiation of mechanical ventilation and expected to be ventilated a further 48 hours or more. Patients will receive usual care or usual care plus two 30 min rehabilitation sessions 5 days/week.

Feasibility outcomes are (1) recruitment of one to two patients per month per site; (2) protocol fidelity with >75% of patients commencing interventions within 72 hours of mechanical ventilation, with >70% interventions delivered; and (3) blinded outcome measures recorded at three time points in >80% of patients. Secondary outcomes are (1) strength and function, the Physical Function ICU Test–scored measured on ICU discharge; (2) hospital length of stay; and (3) mental health and physical ability at 3 months using the WHO Disability Assessment Schedule 2. An economic analysis using hospital health services data reported with an embedded health economic study will collect and assess economic and quality of life data including the Hospital Anxiety and Depression Scales core, the Euroqol-5 Dimension-5 Level and the Impact of Event Score.

**Ethics and dissemination** The study has ethical approval from the South Central Hampshire A Research Ethics Committee (19/SC/0016). All amendments will be approved by this committee. An independent trial monitoring committee is overseeing the study. Results will be made available to critical care survivors, their caregivers, the critical care societies and other researchers.

## Strengths and limitations of this study

► Will investigate the implementation of a protocolised early rehabilitation intervention that is usual care in one NHS/university teaching institution, into other NHS institutions with different organisational structures.

► The defined cohort has been demonstrated to benefit from this type of rehabilitation in alternative healthcare systems.

► Results will inform the design of a multicentre randomised controlled trial.

► This study is not designed to assess the effectiveness of the intervention.

► Inability to blind the intervention to patients, physiotherapist and clinicians involved in the delivery of the intervention.

**Trial registration number** NCT03771014.

## INTRODUCTION

In 2018/2019, there were over 290000 admissions to adult intensive care units (ICUs) in the UK.[1] Treatment advances have reduced mortality associated with critical illness[2 3]; however, survival does not represent the end of the story.[4] A complex interplay between baseline health status, acute disease and the traumatic effects of intensive care treatment is associated with long-term physical, psychological and social hardship.[5–10] Patients discharged from the ICU have higher mortality, higher health service costs and a reduction in employment status compared with hospitalised patients not requiring ICU.[8 11]

ICU-acquired weakness is characterised by rapid muscle wasting, polyneuropathy and bone demineralisation, causing pain, weakness and impaired physical function.[12–14] Contributing factors are multifactorial,

although immobility due to the sedation required for tolerance of ventilation plays an important role.[15 16] Early mobilisation may mitigate these effects.[17–19] In 2009, Schweickert *et al* reported that patients who underwent early physical therapy (within 1.5 days of mechanical ventilation) had greater functional independence at hospital discharge than patients who had usual care physical therapy.[20] A recent randomised controlled trial (RCT) on the impact of a progressive ICU mobility programme reported improved functional status at ICU discharge.[21] Meta-analyses and systematic reviews report that early mobilisation of ICU patients may reduce duration of mechanical ventilation and improve short-term physical outcomes,[22–24] but mobilisation can be difficult to implement during a patient's stay in the ICU. Moreover, studies which used delayed rehabilitation, often more than a week after ICU admission,[25–27] have not replicated these outcomes.[28–34] Barriers to early mobilisation include heavy sedation, patient's illness, lack of resources and/or clinician buy-in.[35–38] In-bed cycle ergometry can provide passive activity in heavily sedated patients who are receiving vasopressors[39 40] with minimal physiological demand[40 41] and can be transitioned to active cycling as the patient's condition improves.[23 42–44]

We implemented cycle ergometry as part of an early protocolised rehabilitation quality improvement programme with physiotherapy technicians supporting the additional workload.[45] Like other investigators, we reported reduced number of ventilator days and ICU length of stay.[21 46–49]

The primary aim of this study was to evaluate the feasibility of an RCT investigating the effect of early protocolised rehabilitation versus usual physiotherapy care in ICU patients. Results will inform a prospective fully powered multicentre RCT. This protocol is reported according to Standard protocol items for clinical trials (SPIRIT 2013 Statement)[50] and Template for Intervention Description and Replication[51] guidelines.

## Aim

The aim of this study was to determine the feasibility of delivering study procedures comparing an early protocolised mobilisation programme that includes cycling with usual care.

## Objectives

Feasibility will be determined by measures of the recruitment process, intervention fidelity and outcome measurement completeness, specifically, (1) study accrual rates: a minimum of 30% of eligible patients or one to two patients per site per month are enrolled; (2) protocol adherence: 75% of patients commencing intervention within 72 hours of ICU admission, with a minimum of 70% of planned interventions delivered; and (3) blinded outcome assessment: functional assessment performed at three time points in 80% of survivors. The results will inform a larger fully powered RCT.

## METHODS AND ANALYSIS
### Study design

This is a two-centre feasibility study using a two-arm RCT, randomised 1:1, with blinded outcome assessments at ICU discharge, hospital discharge and 3-month follow-up. Patients will be recruited from two general ICUs, located in the south of the UK. They will not be recruited from our ICU on account that the intervention is now standard practice at this site. Prior to each site opening to recruitment, an audit of current physiotherapy practice will be undertaken over a 4-week period to evaluate what constitutes 'usual care' in each institution.

### Participants

Ninety patients will be recruited. Eligible patients will be over 42 years old and will have an acute/unplanned medical admission to the ICU. They will be functionally independent prior to ICU admission (Barthel Index>80), in the hospital for <5 days prior to intubation and

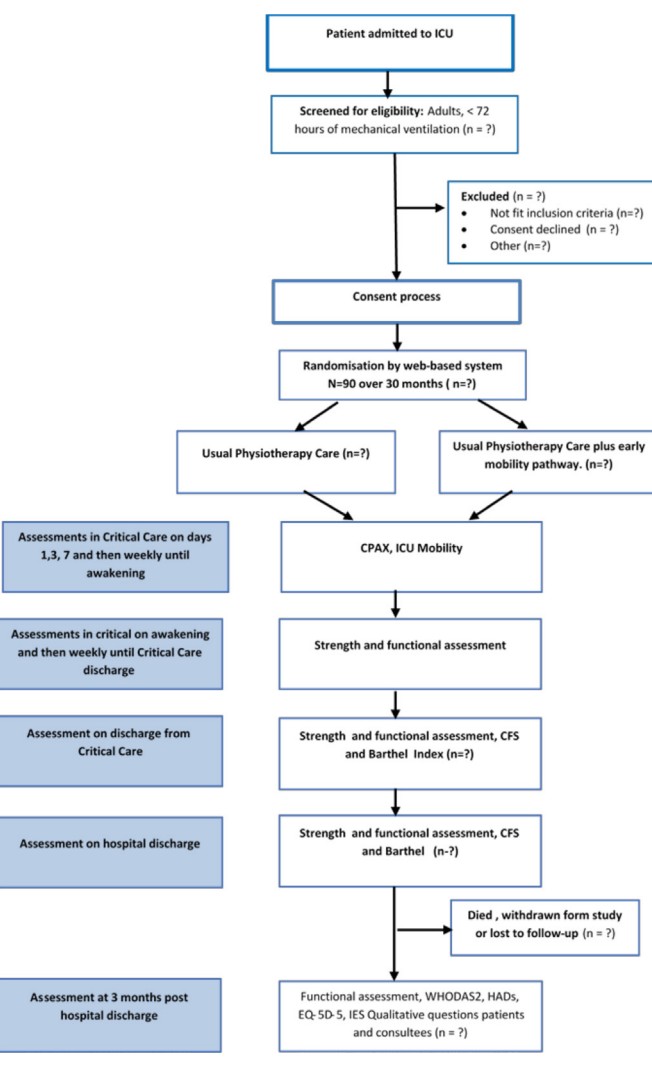

**Figure 1** Study design. CFS, Clinical Frailty Score; CPAx, Chelsea Critical Care Physical Assessment Tool; EQ-5D-5L, Euroqol-5 Dimension-5 Level; HAD, ICU, intensive care unit; IES, Impact of Event Score; WHODAS 2.0, WHO Disability Assessment Schedule 2.

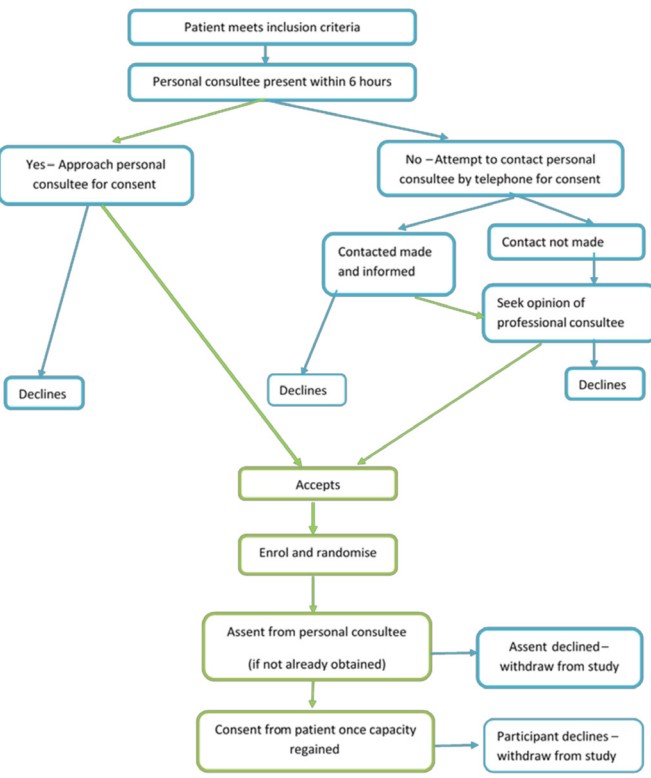

**Figure 2** Consent pathway.

ventilation, intubated and ventilated for <72 hours, and expected to remain ventilated for a further 48 hours. Patients will be excluded if they are in the hospital for 5 days or more prior to ICU admission, have acute brain or spinal cord injury, known or suspected neurological/muscular impairment, condition limiting use of cycle ergometry (eg, lower limb fracture/amputation), not expected to survive >48 hours decided by consulting an intensivist and persistent therapy exemptions in the first 3 days of mechanical ventilation. Figure 1 presents the planned flow of patients through the study.

### Recruitment, consent and randomisation

The study team will screen all patients for eligibility. Recruitment began in June 2019 (and was temporarily suspended in March 2020 due to the COVID-19 pandemic). It is anticipated recruitment will continue until mid 2022. The majority of patients will have diminished capacity when first eligible; therefore, the consent process is multilayered and designed in accordance with the Mental Capacity Act (MCA) 2005[52] (figure 2):

*Patient informed consent*: wherever possible, informed consent will be directly sought from the patient (see online supplemental files 1 and 2).

*Personal consultee informed assent*: if the patient is unable to provide consent, informed assent will be sought from the patient's personal consultee, within 6 hours of confirmation of eligibility. If the personal consultee is not available in person, attempts will be made to contact them by telephone. They will be asked to provide

written assent, at the earliest possible convenience (see online supplemental files 3 and 4).

*Professional consultee informed assent:* where both patient and personal consultee are not available to approve enrolment within 6 hours of confirmation of eligibility, assent will be sought from a professional consultee in accordance with the MCA. The professional consultee will be a consultant medical practitioner, independent from the study. The patient's personal consultee will be consulted at the earliest possible opportunity and assent will be requested to continue in the study.

In all cases, once the patient has regained capacity, they will be informed of the study and consent continuation will be sought. Following consent or assent, patients will be registered on a bespoke electronic data collection tool (ALEA) and randomly assigned to the protocolised early rehabilitation or usual care.

### Staff training/site set-up

Participating sites will employ the equivalent of a full-time therapy technician to deliver the study intervention, under the supervision of a senior critical care therapist. Both senior critical care therapists and therapy technicians will complete a training package delivered by the primary institution (University Hospital Southampton NHS Foundation Trust), where early rehabilitation with cycling is well established and embedded in usual care. This package includes seminars on the delivery of the protocolised early rehabilitation, use of the bespoke electronic database and 5 days of clinical shadowing.

### Interventions

All patients will receive usual medical, nursing and physiotherapy care while in intensive care. Each bedside nurse will be asked at the start of the shift if they have been involved caring for a patient in the intervention arm of the study. The ICU physiotherapy team, who are not involved with the delivery of the study delivery, will deliver all usual physiotherapy interventions in both groups. The physiotherapist delivering usual care will be asked to verify if they have delivered any of the study interventions. In the intervention arm, the protocolised physiotherapy programme will commence within 72 hours of ICU admission or as soon as possible thereafter and will continue for 28 days or until ICU discharge, whichever occurs first. Patients' respiratory support can range from full mandatory ventilation through to oxygen supplementation with no mechanical support following extubation. Sedation is targeted throughout the time that the patient is intubated, and the ventilation mode is adjusted to patients' needs, compliance and comfort at discretion at the start of each physiotherapy intervention, the participants' level of sedation will be assessed using the Richmond Agitation–Sedation Scale (RASS)[53 54] and the Confusion Assessment Method for ICU[55] will be undertaken. RASS will be targeted to a RASS between −1 and +1 by the bedside nurse. After 28 days of ICU admission, all patients will receive usual care physiotherapy interventions.

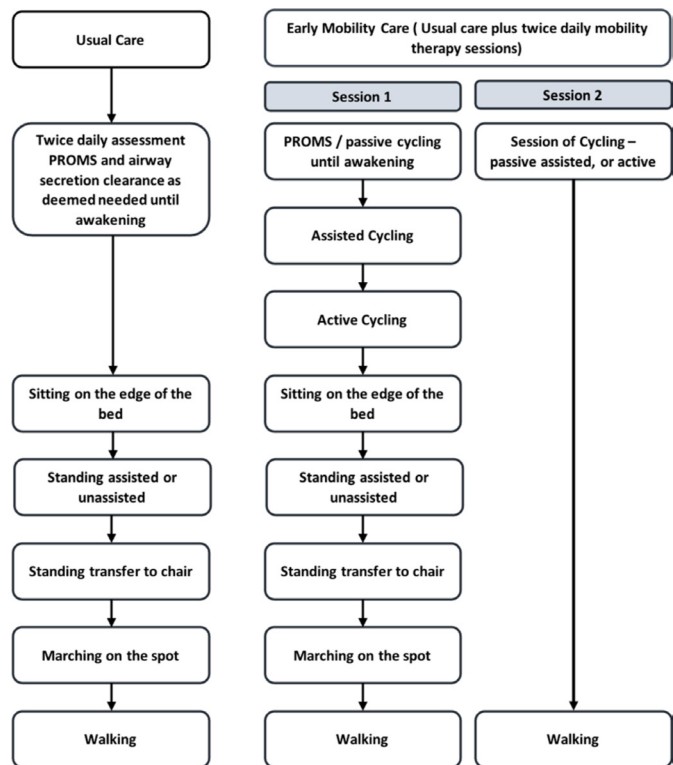

**Figure 3** Study intervention pathway. (PROM = Passive Range of Movement)

## Group 1: usual care control group

In this pragmatic study, physiotherapy interventions will be guided by individual assessment and start in accordance with the usual care pathway within each institution. The focus of each session may be respiratory support, mobilisation or a combination of both. Interventions delivered will be determined by the physiotherapist in conjunction with the attending physician. Interventions include, where appropriate, passive or active range of movement, positioning and respiratory physiotherapy, and when able, sitting on the edge of the bed, standing (assisted or unassisted), standing to transfer to chair, marching on the spot and walking (figure 3). Usual interventions may occur at any time of day.

## Group 2: Protocolised rehabilitation pathway

Patients will have usual care physiotherapy, in addition to the two protocolised intervention within 72 hours of ICU admission or as soon as possible thereafter. Patients will be screened for safety criteria to withhold the intervention prior to each planned intervention session (table 1).

Those meeting criteria to withhold interventions will have issues addressed and reassessed for interventions 2 hours later. The two additional rehabilitation sessions will be delivered by the research physiotherapy staff including a therapy technician. This will comprise two mobility sessions the modality of the first, chosen at the discretion of the physiotherapist. The second session will be 30 min of supine cycling delivered in the afternoon.

The first rehabilitation intervention each day will be delivered in the morning. Planned interventions include passive or active range of movements, passive cycling, active cycling, in-bed exercises, sitting, mobilisation out of bed and walking. Daily assessment of the patient will be made to ensure the highest level of activity possible is provided for each individual patient given safety considerations and capability of the patient.

The second session will be cycling based. An in-bed supine cycle ergometer (MotoMed Letto 2) will be used

| Table 1 | Safety criteria for delivery of physical therapy interventions | |
|---|---|---|
| | **Criteria to commence physiotherapy** | **Criteria to stop/withhold physiotherapy intervention** |
| Blood pressure | Mean Arterial Blood Pressure (MAP) 60–100 mm Hg, no change in vasopressor dose requirement for preceding 2 hours | Catecholamine-resistant hypotension with MAP <60 mm Hg |
| Heart rate | Between 40 and 140 beats/min | <50 or >140 beats/min |
| Respiratory rate | Sustained <40 breaths/min | Sustained >40 breaths/min |
| Temperature | | >40°C |
| Oxygen requirement | If Fraction inspired oxygen ($FiO_2$)>0.8 for passive exercise only | |
| | $FiO_2$ <0.8 and (Positive End Expiratory Pressure) PEEP <15 $cmH_2O$ | |
| Desaturation | | Sats fall <85% for >1 min |
| Other | | ▶ Fall.<br>▶ Unplanned extubation.<br>▶ Acute bleeding.<br>▶ New-onset arrhythmia.<br>▶ Signs/symptoms of acute myocardial ischaemia.<br>▶ Patient pain/distress.<br>▶ Clinical team decides therapy intervention not appropriate.<br>▶ Refusal by patient or representative. |

to engage the participant in passive, assisted or active cycling, or a combination, depending on the degree of patient cooperation (figure 3). The aim was for the patient to have 30 min of cycling per day, following a standardised cycling programme. If cycling is in passive mode, patients will commence cycling at 5 revolutions per minute (RPM), building up to 20 RPM over 5 min and continue this for 20 min before 5 min 5 RPM cool down. In the assisted or active mode, after the 5 min warm-up, cycling will continue for 20 min at patient-selected RPM followed by a 5 min cool-down at 5 RPM. In-bed cycling sessions will stop when the patient is deemed to be able to stand and transfer from bed to chair for both mobility sessions for two consecutive days. If patients are considered unable to have concurrent mobility therapy and respiratory weaning, mobility therapy will take priority, in agreement with the senior clinical team. Individual participants will receive the trial intervention on 5 days/week (Monday–Friday) for the duration of their ICU stay or a maximum of 28 days, whichever comes first. Patients will be monitored for cardiovascular and respiratory stability and safety of indwelling lines, tubes and catheters with predetermined criteria for termination of any session (table 1). Deviations from the planned protocol will be reported to determine potential barriers to implementation. Patients will be able to decline any intervention or outcome assessment at any time without compromise to their care.

## Primary outcome: feasibility to deliver the protocol as designed

Feasibility will be determined by measures of the recruitment process, intervention fidelity and outcome measurement completeness, specifically,

► Study accrual rates: a minimum of 30% of eligible patients or one to two patients per site per month are enrolled.
► Protocol adherence: 75% of patients commencing intervention within 72 hours of ICU admission, minimum of 70% of planned interventions delivered.
► Blinded outcome assessment: functional assessment performed at three time points in 80% of survivors by physiotherapists working within the hospital but not within the ICU.

## Secondary outcomes

The schedule of outcome assessments is detailed in table 2.

## Strength and function

We will measure the Physical Function ICU Test–scored (PFITs) at awakening as described by De Jonghe et al[56]

| Table 2 | Schedule of assessments | | | | | | | | |
|---|---|---|---|---|---|---|---|---|---|
| | Randomisation | Day 1 | Day 3 | Day 7 | Awakening | Weekly | ICU discharge | Hospital discharge | 3 months posthospital discharge |
| Demographic data | X | | | | | | | | |
| Muscle assessment | | | | | | | | | |
| Medicial Research Council sum-score (MRCss)[60 61] | | | | | X | X | X | X | |
| Grip strength[62] | | | | | X | X | X | X | |
| Physical function | | | | | | | | | |
| CPAx[63] | | X | X | X | X | X | X | | |
| ICU mobility[64] | | X | X | X | X | X | X | | |
| PFITs[59] | | | | | X | X | X | | |
| Timed-Up and Go | | | | | | | X | X | X |
| Clinical Frailty Score[69] | | (X) | | | | | X | X | X |
| Barthel Index | | (X) | | | | | X | X | X |
| 6 min Walk Test[70] | | | | | | | | X | X |
| Health Related Quality of LIfe (HRQL) | | | | | | | | | |
| WHODAS 2.0[71] | | | | | | | | | X |
| HADS[72 73] | | | | | | | | | X |
| EQ-5D-5L[74] | | | | | | | | | X |
| Impact of Event Scale[75] | | | | | | | | | X |
| Health Economic Evaluation (CSRI)* | | | | | | | | | X |

CPAx, Chelsea Critical Care Physical Assessment Tool; CSRI, Client Service Receipt Inventory; EQ-5D-5L, Euroqol-5 Dimension-5 Level; HADS, Hospital Anxiety and Depression Scale; ICU, intensive care unit; PFITs, Physical Function ICU Test–scored; WHODAS 2.0, WHO Disability Assessment Schedule 2.

then weekly within ICU and on ICU discharge.[57] PFITs is a reliable and valid four-item scale (arm strength, leg strength, ability to stand and step cadence), with a score range of 0–10 and is responsive to change and predictive of key outcomes.[58] Medical Research Council Manual Muscle Test Sum Score (MRC-ss)[59 60] and handheld dynamometry[61] will be measured on awakening, weekly, on ICU discharge and hospital discharge. Chelsea Critical Care Physical Assessment Tool (CPAx)[62] and ICU Mobility Scale[63] will be assessed three times during the first week within ICU, on awakening, weekly thereafter within the ICU and at ICU discharge. Timed Up and Go,[64 65] Clinical Frailty Score (CFS)[66–68] and Barthel Index will be assessed at ICU discharge, hospital discharge and 3 months posthospital discharge. Preadmission Barthel Index and CFS will be assessed by proxy on admission from family member or next of kin. Six-minute walk test[69] will be performed, in accordance with American Thoracic Society guidelines, at hospital discharge and 3 months posthospital discharge.

### Health-related quality of life (QOL) outcomes

The following will be measured at 3 months posthospital discharge : WHODAS 2.0,[70] Hospital Anxiety and Depression Scale score,[71 72] Euroqol-5 Dimension-5 Level (EQ-5D-5L),[73] Impact of Event Score[74] and Client Service Receipt Inventory questionnaire, designed for this study to evaluate costs that fall on patients and their carers. Resource use and costs including direct intervention costs of therapists and equipment and general hospital costs (per bed day) will be recorded for each patient.

### Health economic substudy

We will also conduct an embedded health economic study to identify and define data collection for a future RCT where a full cost-effectiveness analysis (CEA) can be conducted. Within the feasibility study, we aim to address the following:

► What the quality of the data and what potential problems are there for reporting QoL (EQ-5D-5L), resource use and costs.
► The cost implications of the proposed intervention in terms of impact for the NHS (inpatient stay bed days) and identifying the main cost drivers.
► Is the EQ-5D-5L appropriate for use in the future RCT?

The economic outcomes will include secondary care resource use within hospitals during inpatient stay, primary care resource use following discharge up to 3 months and resource use related to providing the intervention. The results will be reported in the form of descriptive statistics and will be used to inform a future CEA within a definitive RCT.

### Additional data collection

We will collect baseline data including demographic information, Functional Comorbidity Index, ICU diagnosis, APACHE II score, ventilation duration, ventilator-free days, ICU and hospital length of stay, within ICU drug history and duration and type of usual care physiotherapy.

### Implementation evaluation

We aim to investigate whether the protocolised early rehabilitation programme used in one NHS institution is transferable, as an RCT, into other similar NHS institutions. The design of a future multicentre study will be informed by identified facilitators and barriers to implementation. Implementation assessment will be based on the measures described by Proctor.[75] A cross section of ICU staff and patients will be interviewed and complete questionnaires at trial completion to identify barriers impacting delivery of the study. Understanding of the integration and sustainability of the intervention are necessary to inform the design of a powered RCT. Acceptability will be measured at the beginning and end of the study from investigators and clinical staff by direct discussions and questionnaire. Our experience informs us that the introduction of this intervention is dependent on a cultural change within any unit for a proactive focus on early mobilisation. We aim to explore measures to help optimise implementation. Adoption, feasibility and fidelity measures will be monitored during the study by regular meetings with the investigators. Patient screening logs will identify the number of patients eligible for the study and barriers to enrolment. We will assess the degree to which it is possible to separate the staff caring for the intervention group from those caring for the patients in the control group.

We will report whether trial participation has influenced usual care within the participating units by prestudy and poststudy audits. Participating sites will collect data regarding number and seniority of therapy staff with dedicated time to work within the ICU; delirium and sedation protocols used; time, type and frequency of rehabilitation interventions delivered, who delivers the interventions and reasons why usual care may not be delivered.

The feasibility outcomes described earlier will be used to power a larger RCT.

### Data entry and checks

Data will be entered into the secure electronic case report form (ALEA) and data validation will take place according to the procedures set out in the data management plan and data validation plan, both developed apriori. Missing data will be assessed to identify any specific challenges with any items of data collected. Missing data level is expected to be less than 20%. Data loss and mortality will inform number of participants needed to design a larger RCT. As this is a feasibility study data imputation will not be undertaken. Prior to statistical analysis, variables will be checked for missing and impossible and improbable values as defined by clinical opinion. Questions regarding the data will be directed to the data manager.

## Sample size calculation

This is a feasibility study, the results of which will be used to power a definitive study if appropriate; as such, no formal sample size calculation for effectiveness of the intervention has been undertaken. Ninety patients will be recruited, aiming for 30–45 participants at each site. We anticipate a 30% in hospital mortality /loss to follow-up with an estimate of 60 patients completing the study. This sample size of 90 will allow the estimate of recruitment rate to be made with a 95% CI of ±5.2% if the rate is observed to be around 30%, and with a CI of ±7.3% if the recruitment rate is observed to be around 50%. In addition, the sample of 90 recruited patients will allow the estimate of the mortality rate to be made with a 95% CI of ±9.5%, assuming the mortality rate was around 30%. Finally, assuming the recruitment rate was around 30%, a sample of 300 patients approached to take part in the study, leading to 90 enrolled patients would allow for the recruitment rate to be estimated with a 95% CI of ±5.2%. If the recruitment rate was nearer 50%, with 180 patients approached to recruit the 90 enrolled patients, the recruitment rate would be estimated with a 95% CI of ±7.3%.

## Statistical analysis

The analysis will be reported in line with the feasibility studies extension to the Consolidated Standards of Reporting Trials statement.[76] The aims of the study were to estimate the recruitment, compliance and retention rates to inform the design of a future study and is not powered for hypothesis testing regarding the effectiveness of the intervention. Feasibility outcomes (recruitment, compliance and retention rates) will be presented with 95% CIs across the whole study population. Compliance and retention rates will also be presented by treatment arm to ensure balanced recruitment, but no formal statistical comparison tests will be made. Mortality and participant dropout rates will be presented with 95% CIs across the whole study population and within treatment arm. Clinical outcome data (secondary outcomes) will be presented as summary statistics using means and SDs or medians and ranges/IQRs, as applicable, across the whole study population and by treatment arm. These data will be used to inform the future trial but will not be used to draw conclusions about the effectiveness of the protocolised early rehabilitation intervention within this study.

## Trial management

The chief investigator will ensure all study personnel are appropriately orientated and trained, oversee recruitment and report to the trial safety monitoring committee. Training will occur across sites using competency-based training developed at the primary site (University Hospital Southampton NHS Foundation Trust). A study steering group, consisting of an independent chair, expert members and two lay advisors will meet every 3 months. Fortnightly teleconferences with trial sites will be held to monitor conduct and progress. Timing and intervals of visits and teleconferences will be reviewed at 3 months to ensure optimal time use.

The chief investigator and principal investigators will facilitate local monitoring by the research and development quality manager, research ethics committee (REC) review and provide access to source data as required. A monitoring report will be produced, summarising the visit, documents and findings. The chief investigator will ensure that all findings are addressed appropriately. The steering group will review all events in a timely manner. Additional monitoring will be scheduled where there is evidence or suspicion of non-compliance with the study protocol.

A data management and safety committee will be chaired by an independent expert. Quarterly reports will be given to the committee once recruitment has commenced.

## Patient and public involvement

The study has been supported by patient advisory representatives. These representatives are members of the trial steering committee. Patient advisors partnered with us for the design of the study, the informational material to support the intervention, the burden of the intervention from the patient's perspective and contributed to the dissemination plan

## Ethics and dissemination

Ethical approval has been granted by South Central Hampshire A Research Ethics Committee (REC reference 19/SC/0016). This study entitled: A feasibility study of Early Mobilisation Programmes in Critical Care (EMPRESS) was registered with Clinical Ttrials.gov on 10 December 2018.

Results of this proposed feasibility study will be disseminated for four key audiences: (1) patients and public; (2) intensive care staff, healthcare workers and potential future research delivery partners; (3) service delivery organisations; and (4) academic and potential future research collaborators. Dissemination activities will include feedback to patients and public involvement study focus group, feedback to study participants, presentations to local clinical teams and managers and commissioners and presentation at conferences attended by appropriate healthcare professionals. Where appropriate, results will be published in peer reviewed journals.

## Safety and adverse events

Early mobility within ICUs is safe. In a review of physiotherapy in a critical care rehabilitation programme, 1110 patients underwent 5267 rehabilitation sessions; physiological abnormalities or potential adverse events occurred in only 6 per 1000 interventions.[77] Mobilisation interventions will only be delivered if patients fit the safety criteria defined in table 1. Similar safety criteria have been used in other ICU rehabilitation studies.[78 79]

All adverse events will be documented. Any intervention will cease according to stopping criteria detailed in

table 1. Any such event will be recorded as an adverse event. The chief investigator will provide a monthly update to the safety monitoring committee. Serious adverse events are events that result in death, are life-threatening or require prolonged hospitalisation. Any such event will be reported in accordance with the NHS Health Research Authority guidance.

## DISCUSSION

EMPRESS is a feasibility study to assess if an RCT of protocolised rehabilitation with supine cycling can be delivered in ventilated patients in ICUs with differing organisational structures with blinded follow-up assessments. A recent meta-analysis indicated that protocolised rehabilitation significantly reduces duration of mechanical ventilation and ICU length of stay.[23] This is consistent with our findings when we introduced the early rehabilitation programme outlined here in our ICU.[45] Passive cycling commenced on ventilated patients may assist the recovery muscle strength in ICU patients,[43] although the overall benefits of leg cycle ergometry in the critically ill is inconclusive.[44] We describe a protocolised rehabilitation programme with supine cycling delivered as close to intubation as possible, at an intensity according to the patients' highest performance capability.

Both patient and organisational issues are recognised to the delivery of early rehabilitation of the critically ill patients.[35] A frequently reported challenge is the lack of appropriately qualified staff.[80] This study evaluates the safety, feasibility, effectiveness of delivery and cost efficiency of using therapy technicians to deliver protocolised rehabilitation interventions. In addition to the clinical benefits, early physical rehabilitation can also be cost saving.[49] Even with the cost of employment of additional therapy technicians specifically to assist in the delivery of we have found this early rehabilitation programme cost effective.[81]

This study will collect data on the dose of intervention delivered to all patients, reasons for non-delivery of protocol interventions, and the level of experience of therapists delivering the interventions. A qualitative process evaluation is designed to identify both patient and organisational challenges that have potential to be addressed in a potential future study. Findings will inform refinement of trial design and evaluation of the intervention, clarifying causal mechanisms behind study outcomes and providing additional context not adequately captured by the quantitative data. The process evaluation will be consistent with Medical Research Council guidance for conducting process evaluations of complex healthcare interventions.[82]

Targeted sedation is embedded within this protocol as oversedation is one of the more commonly cited barriers to mobilisation of the ventilated patient.[35] This study opened to recruitment prior to the publication of the recommended core outcome set for critical care ventilation trials[83]; however, three of the six outcomes listed (duration of mechanical ventilation, duration of stay and health-related QOL) are secondary outcomes in this study and the other three outcomes are included in the data collected. This will be addressed should we proceed to a full RCT. Due to the nature of the intervention, it is not possible for this to be blinded; however, the follow-up assessments will be carried out by a blinded.

Results from EMPRESS will inform the design of a multi-centred RCT, both identifying barriers to the implementation of the designed protocol and exploring how these may be addressed from feedback from the therapy and nursing teams in addition to the feedback from patients and their next of kin.

**Author affiliations**
[1]NIHR Biomedical Research Centre, University Hospital Southampton NHS Foundation Trust, Southampton, UK
[2]Department of Intensive Care, University Hospital Southampton NHS Foundation Trust, Southampton, UK
[3]Department of Physiotherapy, University Hospital Southampton NHS Foundation Trust, Southampton, UK
[4]Melbourne School of Health Sciences, The University of Melbourne, Melbourne, Victoria, Australia
[5]Peter MacCallum Cancer Institute, Melbourne, Victoria, Australia
[6]Guy's and St Thomas' NHS Foundation Trust, London, UK
[7]Respiratory and Critical Care, King's College London, London, UK
[8]Faculty of Medicine, University of Southampton, Southampton, UK
[9]University Hospital Southampton NHS Foundation Trust, Southampton, UK

**Contributors** RC and AB contributed equally to the preparation of the paper. RC and ZvW had the original idea for the study. RC, LD, IR, NH, AD, GS, ID and MG developed the trial protocol. IR devised the statistical analysis plan. MC developed the economic analysis. AB, GS, ID and RC prepared and submitted documents for research and development and ethical approval. RC, KM and AB wrote the manuscript. All authors reviewed the final version.

**Funding** This work is supported by the National Institute for Health Research (NIHR) under its Research for Patient Benefit (RfPB) Programme. Grant Reference Number PB-PG-0317-20045. KM is an NIHR Senior Nurse and Midwife Research Leader. Sponsor: University Hospital Southampton NHS Foundation Trust.

**Disclaimer** The views expressed are those of the author(s) and not necessarily those of the National Institute for Health Research, NHS or the Department of Health and Social Care.

**Competing interests** None declared.

**Patient and public involvement** Patients and/or the public were involved in the design, conduct, reporting or dissemination plans of this research. Refer to the Methods and analysis section for further details.

**Patient consent for publication** Not applicable.

**Provenance and peer review** Not commissioned; externally peer reviewed.

**ORCID iDs**

Rebecca Cusack http://orcid.org/0000-0003-2863-2870
Andrew Bates http://orcid.org/0000-0002-3614-0270
Kay Mitchell http://orcid.org/0000-0001-6393-8475
Linda Denehy http://orcid.org/0000-0002-2926-8436
Nicholas Hart http://orcid.org/0000-0002-6863-585X
Ahilanandan Dushianthan http://orcid.org/0000-0002-0165-3359
Isabel Reading http://orcid.org/0000-0002-1457-6532
Michael Grocott http://orcid.org/0000-0002-9484-7581

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
