## [Reviewer comments · BMJ Open]

ARTICLE DETAILS

TITLE (PROVISIONAL)	Improving physical function of patients following Intensive Care Unit admission (EMPRESS): Protocol of a randomised controlled feasibility trial
AUTHORS	Cusack, Rebecca; Bates, Andrew; Mitchell, Kay; van Willigen, Zoe; Denehy, Linda; Hart, Nicholas; Dushianthan, Ahilanandan; Reading, Isabel; Chorooglou, Maria; Sturme, Gordon; Davey, Iain; Grocott, Michael

VERSION 1 – REVIEW

REVIEWER	Van Aswegen, Helena University of the Witwatersrand, Physiotherapy
REVIEW RETURNED	11-Oct-2021

GENERAL COMMENTS	Thank you for the opportunity to review this protocol that addresses an important aspect of rehabilitation of patients with prolonged stay in ICU to better their immediate and long term clinical outcomes. Generally the protocol is written well, however, there are several sentence construction errors in the Introduction section that should be corrected. The abstract does not include all the QOL assessment tools that will be used. For the sake of transparency these should be mentioned. The sample size of 90 participants for this feasibility RCT needs to be explained further. How was this number decided on? The intervention that is different between the two groups is the addition of passive/active cycling. Otherwise the mobilisation pathways are similar between groups. I suggest that the key words be adjusted accordingly to include cycling and to remove early mobilisation. It is not clear in the Methods whether this feasibility RCT will be done at both Southampton NHS hospital or just at the other study site. If at both, is it ethical to withhold cycling from participants in the control group at Southampton hospital if this forms part of the usual care received by patients admitted to this hospital? The intervention group will receive two treatment sessions daily vs the control group who will receive one treatment session per day. How will the bias that this introduces be addressed in the analysis of findings in the larger RCT? The purpose of including participant assessment with the ICU Mobility scale over and above the CPAX tool and PFIT-s is not clear. This scale is likely to give similar information that one would receive by doing the CPAX tool and PFIT-s only. Would the use of 3 functional assessment tools not be tiring for the participants? Please clarify who will be doing the blinded outcome assessments at ICU discharge, hospital discharge, and at 3 months.
--

REVIEWER	Battaglini, Denise San Martino Policlinico Hospital, IRCCS for Oncology and Neurosciences, Anesthesia and Intensive Care
REVIEW RETURNED	21-Oct-2021

GENERAL COMMENTS	“The objective of the study is to determine the feasibility of recruitment and delivery of a randomised clinical trial comparing an early mobilisation programme including cycling with usual care to inform a larger multicentred study.” At first glance the objective is difficult to understand in the abstract. “Patients will receive usual care or usual care plus two 30- minute rehabilitation sessions 5 days per week.” The timing is also important, it is not clear here. Not clear what you mean for “feasibility outcomes”. The aims should be better specified (first and secondary). Strengths and limitations: “Will investigate the implementation of an early mobilisation intervention” this is the clearest endpoint till now. Introduction: I found the informations too focalized to UK only. I suggest focusing on general situations instead of only UK. Introduction: too long and too descriptive. You should go to the point using few words and examples but focusing more on the main objectives. Methods: it is not clear to me the timing of physiotherapy starting and the modality (i.e., only passive, active). Moreover, it is not clear if you intend only general physiotherapy or respiratory physiotherapy as well. I suggest also adding and explaining each maneuver. Is the patient still intubated/ extubated during physiotherapy? Not clear, very important and should be better explained. Which kind of ventilation has the patient? Assisted? Controlled? Important for the protocol. “The first mobilisation intervention each day will include activities such as PROMS, passive cycling, active cycling, in bed exercises, sitting, mobilisation out of bed and walking”. I strongly suggest protocolizing the interventions (not “such as” but... “the planned interventions are”). I suggest moving the outcomes as first thing in the methods, otherwise not clear for the reader. Statistical analysis is too general. You should specify the tests that you are planning to use for your outcomes. Discussion: too short, here you can use some points from the introduction in order to contextualize better. Also, as I previously commented, the introduction is too long. I would suggest organizing as: 1) what you expect from this study 2) reason about your expectations by citing the appropriate literature. Missing: limitations
--

VERSION 1 – AUTHOR RESPONSE

Reviewer: 1

Dr. Helena Van Aswegen, University of the Witwatersrand

Comments to the Author:

Thank you for the opportunity to review this protocol that addresses an important aspect of rehabilitation of patients with prolonged stay in ICU to better their immediate and long term clinical outcomes.

Comment 1: Generally the protocol is written well, however, there are several sentence construction errors in the Introduction section that should be corrected.

Author's Response: Thank you. The whole of the introduction has been significantly revised and shortened as suggested by reviewer 2 Dr Battaglini. With attention to construction of the new introduction

Comment 2: The abstract does not include all the QOL assessment tools that will be used. For the sake of transparency these should be mentioned.

Author's Response: All of the QOL assessment tools that will be used are now listed to include:(Lines 19-21)

WHODAS 2

Hospital Anxiety and Depression score (HADS)

Euroqol-5 Dimension-5Level (EQ-5D-5L)

Impact of Event Score (IES)

Comment 3:The sample size of 90 participants for this feasibility RCT needs to be explained further. How was this number decided on?

Author's Response: The following reasoning for the sample size of 90 participants has been inserted as below-(Lines 287-300)

This is a feasibility study the results of which will be used to power a definitive study if appropriate, as such no formal sample size calculation for effectiveness of the intervention has been undertaken. 90 patients will be recruited aiming for 30-45 participants at each site. We anticipate a 30% in hospital mortality /loss to follow-up with an estimate of 60 patients completing the study. This sample size of 90 will allow the estimate of recruitment rate to be made with a 95% confidence interval of $\pm 5.2\%$ if the rate is observed to be around 30%, and with a confidence interval of $\pm 7.3\%$ if the recruitment rate is observed to be around 50%. In addition, the sample of 90 recruited patients will allow the estimate of the mortality rate to be made with a 95% confidence interval of $\pm 9.5\%$ assuming the mortality rate was around 30%. Finally, assuming the recruitment rate was around 30%, a sample of 300 patients approached to take part in the study leading to 90 enrolled patients would allow for the recruitment rate to be estimated with a 95% confidence interval of $\pm 5.2\%$. If the recruitment rate was nearer 50%, with 180 patients approached to recruit the 90 enrolled patients, the recruitment rate would be estimated with a 95% confidence interval of $\pm 7.3\%$.

Comment 4: The intervention that is different between the two groups is the addition of passive/active cycling. Otherwise the mobilisation pathways are similar between groups. I suggest that the key words be adjusted accordingly to include cycling and to remove early mobilisation.

Author's Response: Thank you for this comment. It is apparent that the description needs better clarity. The control arm is the usual physiotherapy intervention at the given institution. The intervention arm is the usual physiotherapy plus 2 additional sessions. The first in the morning aiming for the maximum level of mobilisation. The modality is at the discretion of the attending physiotherapist. If the patient is fully sedated, as for example on the day of admission it is likely passive interventions will be used (PROMs and passive cycling). The second afternoon session will comprise of 30minutes cycling.

On the title page the key words 'early ambulation' have been replaced by the word 'cycling'.

The description of the usual care arm has been expanded and the following included (lines 162-165):

In this pragmatic study physiotherapy interventions will start in accordance with the usual care pathway within each institution. The focus of each session may be respiratory support, mobilization or a combination of both. Interventions delivered will be determined by the attending physiotherapist in conjunction with the attending physician as is the usual practice in the UK

The first paragraph of the description of the intervention arm has been amended (lines 168-185):

Participants will receive usual care physiotherapy, in addition to the two protocolised interventions within 72 hours of ICU admission or as soon as possible thereafter. Participants will be screened for criteria to withhold the intervention prior to each planned intervention session (Table 1)

Those meeting criteria to withhold interventions will have issues addressed and will be reassessed for interventions 2 hours later. The two additional mobilisation sessions will be delivered by the research physiotherapy staff including a therapy technician. This will comprise of two mobility sessions the modality of the first, chosen at the discretion of the physiotherapist. The second session will be of 30-minute of supine cycling, delivered in the afternoon.

The first mobilisation intervention each day will be delivered in the morning. Planned interventions include passive or active range of movements, passive cycling, active cycling, in bed exercises, sitting, mobilisation out of bed and walking. Daily assessment of the patient will be made to ensure the highest level of activity possible is provided for each individual patient given safety considerations and capability of the patient.

Comment 5: It is not clear in the Methods whether this feasibility RCT will be done at both Southampton NHS hospital or just at the other study site. If at both, is it ethical to withhold cycling from participants in the control group at Southampton hospital if this forms part of the usual care received by patients admitted to this hospital?

Author's Response: Participant recruitment will not be undertaken at Southampton NHS hospital and the following sentence inserted to reflect this. (Lines 92-93)

Participants will be recruited from two general intensive care units, located in the south of the UK. They will not be recruited from our ICU on account that the intervention is now standard practice at this site

Comment 6: The intervention group will receive two treatment sessions daily vs the control group who will receive one treatment session per day. How will the bias that this introduces be addressed in the analysis of findings in the larger RCT?

Author's response: Thank you this comment. This is a tricky point as the 2 aspects of the intervention (early intervention and 2 additional daily sessions) are linked. In the definitive trial we would try to look at the effect of timing of starting the physiotherapy interventions and separating this from the trajectory of improvement. This feasibility study will give us some data to look at to see if the trajectory of improvement and timing of starting physiotherapy may be able to be teased apart. This will be fully discussed in report of the study when we may have been able to evaluate the impact of this issue.

Comment 7: The purpose of including participant assessment with the ICU Mobility scale over and above the CPax tool and PFIT-s is not clear. This scale is likely to give similar information that one would receive by doing the CPax tool and PFIT-s only. Would the use of 3 functional assessment tools not be tiring for the participants?

Author's response. Thank you for this comment. I fully acknowledge that the ICU mobility scale is likely to add little over and above the CPax and PFIT-s. The reasoning for including the ICU mobility scale is that not all units are familiar with the CPax tool and one of our uncertainties was as to what extent that it would be completed. If the CPax is completed the ICU mobility scale needs little additional input from the patient and indeed so far, the participants have not reported completing 3 assessments too tiring. If the CPax tool completion is incomplete the ICU mobility scale would give us some important information on progress. Further this approach may inform us as to which measures may have best uptake in a definitive trial. The approved protocol includes both of these outcome measures, and the study now has been recruited more than a third of target as such the study team feel to change this at this stage would not be appropriate.

Comment 8: Please clarify who will be doing the blinded outcome assessments at ICU discharge, hospital discharge, and at 3 months.

Author's response: (Lines 205-206) The sentence describing the third feasibility outcome has been extended to include who will be performing the blinded assessments

Blinded outcome assessment: functional assessment performed at 3 time-points in 80% of survivors by physiotherapists working within the hospital but not within the ICU

Reviewer: 2

Dr. Denise Battaglini, San Martino Policlinico Hospital, IRCCS for Oncology and Neurosciences
Comments to the Author:

Comment 1: "The aim of the study is to determine the feasibility of recruitment and delivery of a randomised clinical trial comparing an early mobilisation programme including cycling with usual care to inform a larger multicentred study." At first glance the aim is difficult to understand in the abstract.

Author's response: The objective of the study as stated in the abstract has been rephrased and now reads with appropriate use. (Lines 7-9)

The aim of the study is to determine the feasibility of delivering the designed protocol of a randomised clinical trial, comparing an early mobilisation programme including cycling with usual care. This feasibility study will inform a larger multicentred study

Comment 2: "Patients will receive usual care or usual care plus two 30- minute rehabilitation sessions 5 days per week." The timing is also important, it is not clear here.

Author's response: Due to the pragmatic nature of this study exact timings of the interventions are not prescribed. Usual care interventions are delivered throughout the day as the physiotherapy staff see individual patients in turn and depending on staff availability. The timing of the intervention sessions are delineated such that the first mobilisation session is delivered in the morning and the second supine cycling based is delivered in the afternoon. The description of the timing of these interventions has been rewritten to improve clarity (lines 176-181)

Comment 3: Not clear what you mean for "feasibility outcomes". The aims should be better specified (first and secondary).

Authors response: The wording of the outcomes has been changed and now reads (lines 204-215)):

Primary Outcome: Feasibility to deliver the protocol as designed

Feasibility will be determined by measures of the recruitment process, intervention fidelity and outcome measurement completeness, specifically:

1. *Study accrual rates: a minimum of 30% of eligible patients or 1-2 patients per site per month are enrolled*

2. *Protocol adherence: 75% of patients commencing intervention within 72 hours of ICU admission; minimum of 70% of planned interventions delivered*
3. *Blinded outcome assessment: functional assessment performed at 3 time-points in 80% of survivors by physiotherapists working within the hospital but not within the ICU*

Secondary Outcomes:

The schedule of outcome assessments is detailed in Table 2

Comment 4: Strengths and limitations:

“Will investigate the implementation of an early mobilisation intervention” this is the clearest endpoint till now.

Introduction: I found the informations too focalized to UK only. I suggest focusing on general situations instead of only UK.

Introduction: too long and too descriptive. You should go to the point using few words and examples but focusing more on the main objectives.

Author’s response: Thank you for this comment. The introduction has been considerably shortened and we believe is more focused on the situations generally seen (lines 45-76).

Comment 5: Methods: it is not clear to me the timing of physiotherapy starting and the modality (i.e., only passive, active). Moreover, it is not clear if you intend only general physiotherapy or respiratory physiotherapy as well. I suggest also adding and explaining each manoeuvre.

Authors response: The timing of the physiotherapy interventions has been rewritten – please refer comment 2

Comment 6: Is the patient still intubated/ extubated during physiotherapy? Not clear, very important and should be better explained. Which kind of ventilation has the patient? Assisted? Controlled? Important for the protocol.

Authors response: The physiotherapy interventions commence as soon as possible after intubation of the patient and continues until the patient leaves ICU. As such patients may be having any form of respiratory support from mandatory ventilation through to simple oxygen support via nasal specs after extubation. Sedation is targeted throughout the time that the patient is intubated and ventilation mode adjusted to patients’ needs, compliance and comfort at discretion now reads (Lines 148-151):

Patients respiratory support can range from full mandatory ventilation through to oxygen supplementation with no mechanical support following extubation. Sedation is targeted throughout the time that the patient is intubated and ventilation mode adjusted to patients’ needs, compliance and comfort at discretion

Comment 7:“The first mobilisation intervention each day will include activities such as PROMS, passive cycling, active cycling, in bed exercises, sitting, mobilisation out of bed and walking”. I strongly suggest protocolizing the interventions (not “such as” but... “the planned interventions are”).

Authors response: The wording has been changed to (Lines 181-185):

‘Planned interventions include passive or active range of movements, passive cycling, active cycling, in bed exercises, sitting, mobilisation out of bed and walking. Daily assessment of the patient will be made to ensure the highest level of activity possible is provided for each individual patient given safety considerations and capability of the patient’.

Comment 8: I suggest moving the outcomes as first thing in the methods, otherwise not clear for the reader.

Authors response: Thank you detailing how improvement in clarity may be achieved. The section is now presented as Aims and Objectives to include the primary outcomes. The section now reads (lines 78-88)

Aim *The aim of this study is to determine the feasibility to deliver study procedures comparing a protocolised mobilisation programme that includes cycling with usual care.*

Objectives *Feasibility will be determined by measures of the recruitment process, intervention fidelity and outcome measurement completeness, specifically: i) Study accrual rates: a minimum of 30% of eligible patients or 1-2 patients per site per month are enrolled; ii) Protocol adherence: 75% of patients commencing intervention within 72 hours of ICU admission; minimum of 70% of planned interventions delivered and iii) Blinded outcome assessment: functional assessment performed at 3 time-points in 80% of survivors. The results will inform a larger fully powered RCT.*

Comment 9 Statistical analysis is too general. You should specify the tests that you are planning to use for your outcomes.

Author's response: The planned statistical analysis has been written in greater detail and now reads (lines 307-320):

The analysis will be reported in line with the feasibility studies extension to the CONSORT statement⁷⁶. The main aims of the study are to estimate the recruitment, compliance and retention rates to inform the design of a future study and is not powered for hypothesis testing regarding the effectiveness of the intervention. Feasibility outcomes (recruitment, compliance, and retention rates) will be presented with 95% confidence intervals across the whole study population. Compliance and retention rates will also be presented by treatment arm to ensure balanced recruitment, but no formal statistical comparison tests will be made. Mortality and participant dropout rates will be presented with 95% confidence intervals across the whole study population and within treatment arm. Clinical outcome data (secondary outcomes) will be presented as summary statistics using means and standard deviations or medians and ranges/interquartile ranges, as applicable, across the whole study population and by treatment arm. These data will be used to inform the future trial but will not be used to draw conclusions about the effectiveness of the early mobilisation intervention within this study

Comment 10: Discussion: too short, here you can use some points from the introduction in order to contextualize better. Also, as I previously commented, the introduction is too long. I would suggest organizing as: 1) what you expect from this study 2) reason about your expectations by citing the appropriate literature.

Authors comment: Thank you for this useful guidance. The discussion has been fully rewritten (Lines 366-400).

VERSION 2 – REVIEW

REVIEWER	Van Aswegen, Helena University of the Witwatersrand, Physiotherapy
REVIEW RETURNED	03-Mar-2022
GENERAL COMMENTS	Thank you for providing the revised manuscript for review. All the concerns that I raised in my previous review have been adequately addressed. The only remaining comment that I have is that the author/s read the manuscript through carefully and correct all the remaining errors with regards to use of punctuation and incomplete sentences.

VERSION 2 – AUTHOR RESPONSE

Reviewer 1

The only remaining comment that I have is that the author/s read the manuscript carefully and correct all the remaining errors with regards to using of punctuation and incomplete sentences. Amendments to punctuation and sentences have been undertaken.

- We note that your trial registration does not currently contain a data sharing statement. BMJ Open adheres to the ICMJE guidelines on the registration of clinical trials, which states: “clinical trials that begin enrolling participants on or after 1 January 2019 must include a data sharing plan in the trial's registration”. See: <http://www.icmje.org/recommendations/browse/publishing-and-editorial-issues/clinical-trial-registration.html>
Please update your registration to include a data sharing statement. The trial registration has now been updated to include a data sharing statement: Immediately following publication individual participant data that underly the results will be de-identified and shared on request. Proposals should be directed to EMPRESS@uhs.nhs.uk